

# A quality assessment algorithm for no-reference images based on transfer learning

Yang Yang, Chang Liu, Hui Wu and Dingguo Yu

College of Media Engineering, Communication University of Zhejiang, Hang Zhou, China

## ABSTRACT

Image quality assessment (IQA) plays a critical role in automatically detecting and correcting defects in images, thereby enhancing the overall performance of image processing and transmission systems. While research on reference-based IQA is well-established, studies on no-reference image IQA remain underdeveloped. In this article, we propose a novel no-reference IQA algorithm based on transfer learning (IQA-NRTL). This algorithm leverages a deep convolutional neural network (CNN) due to its ability to effectively capture multi-scale semantic information features, which are essential for representing the complex visual perception in images. These features are extracted through a visual perception module. Subsequently, an adaptive fusion network integrates these features, and a fully connected regression network correlates the fused semantic information with global semantic information to perform the final quality assessment. Experimental results on authentically distorted datasets (KonIQ-10k, BIQ2021), synthetically distorted datasets (LIVE, TID2013), and an artificial intelligence (AI)-generated content dataset (AGIQA-1K) show that the proposed IQA-NRTL algorithm significantly improves performance compared to mainstream no-reference IQA algorithms, depending on variations in image content and complexity.

## INTRODUCTION

Image quality assessment (IQA) is fundamental in evaluating the perceptual quality of images, playing a critical role in fields such as image processing, computer vision, and digital media (*Cao et al., 2023a*; *Heidari et al., 2024*). Traditional IQA techniques typically rely on reference-based models, where pristine images are available for comparison. However, in many real-world applications, especially with the proliferation of user-generated and artificial intelligence (AI) generated content (AIGC), a reference image is often unavailable. This has increased the demand for no-reference IQA (NR-IQA) algorithms, which can evaluate image quality without any reference. Recent advancements in AI technologies, such as deep learning, have further accelerated the generation of large volumes of digital content, including manipulated media like deepfake images *Heidari et al. (2024)*, heightening the need for accurate and robust NR-IQA methods to ensure content quality and reliability.

Corresponding author
Dingguo Yu, yann@cuz.edu.cn

However, in addition to issues arising from AIGC, the process of image generation, transmission, compression, and storage inevitably introduces various forms of distortion (*Chen et al., 2024*; *Yu et al., 2023*; *Mantiuk, Hammou & Hanji, 2023*). These distortions lead to significant differences between the visual information received by human observers and the original image. Such degradation, if left unaddressed, can result in unexpected deviations in practical applications that rely on high-fidelity image processing, such as medical imaging, autonomous driving, and multimedia communications. Therefore, accurate IQA is essential to mitigate these issues and ensure reliable use of digital images in practice.

Traditionally, full-reference IQA methods, which compare a distorted image with a pristine reference image, have demonstrated significant success in detecting image quality degradation. However, in many real-world scenarios, such as social media sharing or surveillance systems, reference images are not available. As a result, no-reference IQA methods, which evaluate image quality without requiring a reference image, have become a critical area of research. Despite the growing need for these methods, studies on no-reference IQA remain underdeveloped compared to reference-based approaches.

This article addresses the growing need for accurate and efficient no-reference IQA by proposing a novel algorithm based on transfer learning and deep convolutional neural network (CNN). Traditional methods often struggle with distorted images due to data limitations and computational challenges, particularly when assessing AI-generated images with complex visual features. Leveraging transfer learning allows our model to extract meaningful, multi-scale semantic information across diverse image types, enhancing its ability to handle limited training data. The deep CNN, combined with an adaptive fusion network, captures and refines these features to provide a robust and comprehensive assessment of image quality, offering a more accurate and reliable prediction for a wide range of digital images and sources.

## LITERATURE REVIEW

The literature review in this study is divided into two key sections: Image quality assessment (IQA) and no-reference image quality assessment (NR-IQA). The first section provides an overview of the fundamental principles and advancements in IQA, which is a well-established field that plays a crucial role in evaluating the quality of images across various applications. The second section focuses specifically on NR-IQA, a rapidly growing research area that addresses the challenges of assessing image quality without relying on reference images.

### Image quality assessment

IQA, as a critical research direction in the field of computer vision and image processing, plays an essential role across diverse applications such as daily life, medical diagnostics, and military operations (*Zhao et al., 2023*). Accurately assessing image quality allows digital image processing systems to deliver optimal performance, ensuring the reliability of subsequent operations like denoising, enhancement, and feature extraction. Effective IQA not only improves the design and optimization of image processing algorithms but also

provides real-time feedback in practical applications, ultimately enhancing user experience and the efficiency of image-based tasks. In daily life, the demand for ultra-high-definition (UHD) videos presents challenges in balancing efficiency and accuracy during video quality assessment (VQA). For instance, the resolution and complexity of UHD content often necessitate deep learning-based methods to ensure accurate quality evaluation without sacrificing processing speed (*Wu et al., 2023b*). In medical imaging, accurate IQA is vital for ensuring that doctors receive clear and detailed diagnostic information. *Kastryulin et al. (2023)* emphasizes that precise IQA lead to enhanced diagnosis accuracy, enabling more effective and timely treatments. In military applications, where images are often captured in dynamic or challenging environments, high-quality IQA is crucial for the reliability of surveillance systems. *Wenqi et al. (2024)* points out that the clarity and accuracy of such images support intelligence gathering and decision-making processes, underscoring the importance of IQA in defense and security contexts. As technology advances, IQA methods have evolved from traditional approaches based on the human visual system (HVS) to automated techniques leveraging deep learning and CNN. This shift has allowed IQA methods to handle increasingly complex image data more accurately, especially in the face of big data and AIGC. Traditional methods often struggle with assessing the quality of images that lack a reference, such as those encountered in real-time video streams or AI-generated images. No-reference IQA methods, therefore, have gained considerable attention due to their ability to assess image quality without access to a pristine reference image.

Recent advancements in deep learning, particularly in transfer learning, have enabled models to improve their capacity for processing complex and large-scale data. For example, *Chen et al. (2024)* introduced a top-down approach for IQA, where high-level semantic information guides the assessment process, improving the model's ability to focus on semantically important regions. This method, enhanced by cross-scale attention mechanisms, demonstrated notable gains in assessment performance. Similarly, *Zhao et al. (2023)* proposed a self-supervised learning-based approach for blind image quality assessment (BIQA), utilizing custom pre-tasks and quality-aware contrastive loss to enhance model sensitivity to distortions. By leveraging large-scale pre-training on ImageNet, this model achieved significant improvements on subsequent BIQA tasks. *Zhang et al. (2023b)* introduced a multi-task learning scheme that shares model parameters across multiple tasks, such as scene classification and distortion type identification, resulting in enhanced performance on multiple IQA datasets. In addition to these innovations, specialized applications such as face image quality assessment (FIQA) have seen significant advances. *Boutros et al. (2023)* proposed CR-FIQA, a method designed to predict image quality by estimating the relative classifiability of samples, which is particularly effective in assessing the quality of unseen data. *Aslam et al. (2024)* addresses the limitations of IQA by leveraging visual representation learning, achieving state-of-the-art performance on multiple datasets. The model outperforms an ImageNet pre-trained model in Pearson and Spearman correlations, with fewer epochs required. The growing trend of using deep learning-based approaches highlights the importance of integrating

advanced machine learning techniques into IQA, making methods more adaptable to complex real-world scenarios.

Given these advancements, this study aims to build on the current state of no-reference IQA by incorporating transfer learning techniques to address the challenges posed by the increasing prevalence of AI-generated content and complex image distortions. By leveraging CNNs, this approach enhances the ability to capture multi-scale semantic features, making it more robust in handling diverse image content without requiring reference images. As IQA continues to evolve, these no-reference methods will play an increasingly vital role in maintaining the quality of images in various fields.

## No-reference image quality assessment

IQA can be broadly categorized into subjective and objective methods (*Cao et al., 2023b*). Objective IQA methods, in particular, are classified based on the amount of reference image information available: full-reference IQA (FR-IQA) (*Lang et al., 2023*; *Elloumi, Loukil & Bouhlel, 2024*), semi-reference IQA (SR-IQA) (*Liu et al., 2023*), and no-reference IQA (NR-IQA) methods. Among these, NR-IQA (*Bouhamed et al., 2023*; *Zhou et al., 2023*) stands out due to its ability to assess image quality without requiring access to a pristine reference image. This capability makes NR-IQA particularly advantageous for real-time applications and scenarios where obtaining reference images is impractical or impossible.

Traditional NR-IQA methods have relied heavily on the HVS to extract perceptual features from distorted images (*Zhang et al., 2023a*). These methods often train shallow regression models that map distortion-related features to quality scores. While effective in certain contexts, this approach has inherent limitations. The complexity of simulating the HVS's perception process means that these models struggle to capture the full range of distortions that can occur in digital images, especially when it comes to representing subtle structural distortions and intricate visual information. Recent developments in deep learning, particularly CNN, have revolutionized NR-IQA by enabling models to learn more representative features from distorted images. CNN-based approaches utilize multiple convolutional layers to extract hierarchical features, followed by regression networks to predict image quality scores. These methods have shown superior performance compared to traditional NR-IQA approaches, offering more robust and accurate assessments of distorted images (*Han, Liu & Xie, 2023*; *Ruikar & Chaudhury, 2023*). However, CNN-based NR-IQA models face two significant challenges. First, they require extensive training datasets to achieve high accuracy, but current publicly available IQA databases are often too small or insufficiently diverse to support the training needs of these data-hungry models. Second, the computational cost of training deep CNN is substantial, making them less practical for real-time or large-scale IQA tasks (*Pan et al., 2022*).

To address these limitations, transfer learning has emerged as a promising solution in the development of NR-IQA methods. By leveraging pre-trained models from related tasks, transfer learning allows NR-IQA models to achieve high performance with significantly reduced training data and computational resources. This approach mitigates the challenges posed by limited IQA datasets and lengthy training times. In particular,

transfer learning enables the extraction of multi-scale semantic features from images, improving the model's ability to assess image quality across diverse content types and distortion levels. In this study, we propose a novel no-reference IQA algorithm based on transfer learning to overcome the data limitations and computational inefficiencies of traditional CNN-based NR-IQA methods. By leveraging pre-trained CNN models, our approach captures essential visual and semantic features with minimal additional training, resulting in a more efficient and accurate quality assessment. Furthermore, by incorporating adaptive fusion networks, our model integrates multi-scale features more effectively, leading to improved performance across various image content and distortion types. This method not only advances the field of NR-IQA but also makes image quality assessment more feasible in practical, real-time scenarios.

Based on the above content, this article introduces transfer learning into image quality assessment, and innovatively proposes a no-reference image quality assessment algorithm based on transfer learning based on deep convolutional neural network to realize no-reference image quality assessment for distorted images.

### Research gap

Based on the literature, there are notable advancements in IQA, particularly with CNNs and deep learning approaches. However, a clear research gap remains in the domain of NR-IQA. Traditional NR-IQA methods, relying on HVS features, fail to capture the complex distortions in modern image datasets. Although CNN-based methods have improved NR-IQA, they require extensive datasets and computational resources, which limit their practical applicability. Existing IQA databases are often too small to train CNNs effectively, and the time-consuming nature of training deep models makes them unsuitable for real-time image assessment.

Furthermore, while transfer learning has been successfully applied to various image processing tasks, its potential in NR-IQA remains underexplored. Current research lacks robust models that efficiently utilize pre-trained CNN to enhance NR-IQA performance across diverse image types and distortions. Our research aims to address this gap by proposing a novel NR-IQA algorithm based on transfer learning, which overcomes the limitations of traditional methods by reducing training requirements and enhancing efficiency, while delivering superior accuracy in IQA.

## ALGORITHM FLOW AND IMPLEMENTATION STEPS

### Algorithm flow

This article innovatively proposes a quality assessment algorithm for no-reference images based on transfer learning. The algorithm mainly consists of the following four modules: multi-scale semantic information feature extraction module, visual perception module, adaptive fusion network module and quality assessment regression module. Firstly, the distorted images to be assessed are cropped into groups of three image blocks and then input into a deep convolutional neural network based on ResNet-50. This algorithm uses different convolutional layers of ResNet-50 to output local semantic features $fea_1$, $fea_2$, $fea_3$ and high-level semantic features $fea_{high}$. These features are processed by the visual

perception module to obtain local semantic features $fea'_1$, $fea'_2$, $fea'_3$ and $fea'_{high}$ that are more in line with the human visual perception system. Among them, $fea'_1$, $fea'_2$ and $fea'_3$ are processed by the adaptive fusion network module and connected with $fea'_{high}$ to obtain the multi-scale semantic feature $fea_{total}$. Finally, $fea_{total}$ is input into the fully connected layer regression network to predict the image quality score. In order to solve the problem of insufficient training data, the algorithm innovatively adopts transfer learning to transfer and apply the knowledge of the source domain to the training process, avoiding the complex process of training the target network from scratch. The algorithm flow chart is shown in the Fig. 1.

Firstly, the multi-scale semantic information feature extraction module in this algorithm uses a 50-layer residual neural network (ResNet-50) as the backbone network. ResNet-50 has a deep network structure that can effectively capture the semantic information in the images and alleviate the gradient vanishing problem in deep network training through residual connections.

Secondly, the visual perception module in this algorithm is designed to simulate the working principle of the human visual system. It draws on the way the human eyes processes visual information. It can effectively capture the structural and semantic information in the images, and achieve a deep understanding of images information through complex features extraction and representation learning.

Thirdly, this algorithm introduces an adaptive fusion network module, which aims to strengthen the important features of local semantic information of the images while suppressing irrelevant features. The adaptive fusion network learns the weight relationship between images features and realizes adaptive adjustment of different features, thereby more effectively capturing the key information in the images.

Finally, the quality assessment regression module in this algorithm connects the fused local semantic information with the global information. The connected features are input into the fully connected layer regression network to predict the quality score of the images. Through this regression module, the quality assessment of the images can be transformed into a regression problem to obtain the final assessment result.

## Meaning and goal of the algorithm

The significance of the algorithm proposed in this study lies in its innovative approach to tackling the challenges of no-reference image quality assessment. By integrating transfer learning with deep convolutional neural networks, this method focuses on extracting multi-scale semantic features that more accurately simulate human visual perception. This advancement allows the algorithm to assess image quality without relying on reference images, representing a significant improvement over traditional methods. Additionally, the use of transfer learning reduces the need for large datasets, which are typically required for training deep networks, thus making the algorithm more efficient and practical for real-world applications. The goal of the algorithm is to substantially enhance the accuracy and efficiency of no-reference image quality assessment, especially in scenarios where large-scale training data is not available. By employing a multi-layer feature extraction process based on ResNet-50, along with adaptive feature fusion and a regression network, the

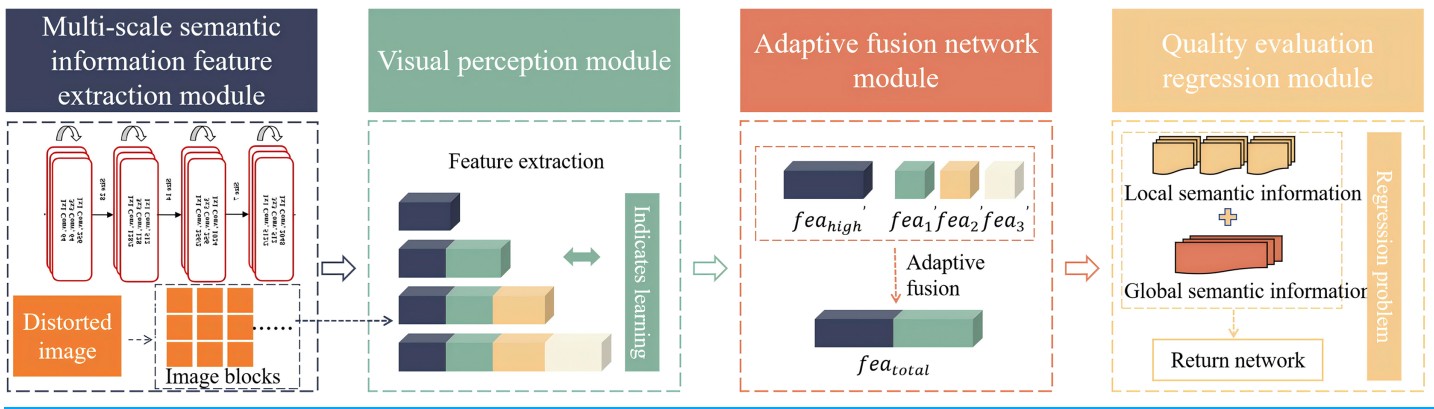

**Figure 1 Algorithm flow.**

algorithm aims to provide more precise quality scores for distorted images. Ultimately, this approach seeks to offer a robust and scalable solution that improves the reliability of image quality assessment across various fields.

## Implementation steps

### Multi-scale semantic information feature extraction module

In the field of deep neural networks, in order to obtain multi-scale semantic features, there have been many mature studies based on the strategy of extracting information from different convolutional layers of images.

AlexNet-NDTL (*Kollem et al., 2023*) is based on an improved AlexNet and uses network-based deep transfer learning to extract features from the dataset. Updated GoogLeNet (*Yang et al., 2023*) replaces the $7 \times 7$ convolution kernel of the first layer with three $3 \times 3$ convolution kernels based on GoogLeNet, adds the ECA attention mechanism to the inception module, and uses a residual network to connect the E-inception module, solving the problem of increased information loss and gradient loss due to increased network depth. Improved ResNet-50 (*Wu et al., 2023a*) introduces the Squeeze-and-Excitation attention mechanism to improve the residual unit of ResNet-50, and combines the Swish function and Ranger optimizer to ensure the effectiveness of feature learning and training, further improving model performance. Improved ResNet-50 has a balanced performance compared with typical recognition algorithms such as AlexNet-NDTL (*Kollem et al., 2023*), Updated GoogLeNet (*Yang et al., 2023*), VGG-16 (*Sharma & Guleria, 2023*), ResNet-18 (*Sunnetci et al., 2023*), and DenseNet-201 (*Salim et al., 2023*). It explicitly constructs an identity mapping. The residual network introduces skip connections, which reduces the possibility of gradient vanishing and gradient exploding during training, and improves the performance of deep networks. The flowchart of feature extraction for multi-scale semantic information is shown in Fig. 2.

The input distorted image information $x$ first passes through the first convolutional layer weight layer to obtain the preliminary mapping function $F(x)$, and then reaches the weight layer based on the second convolutional layer through the linear rectification

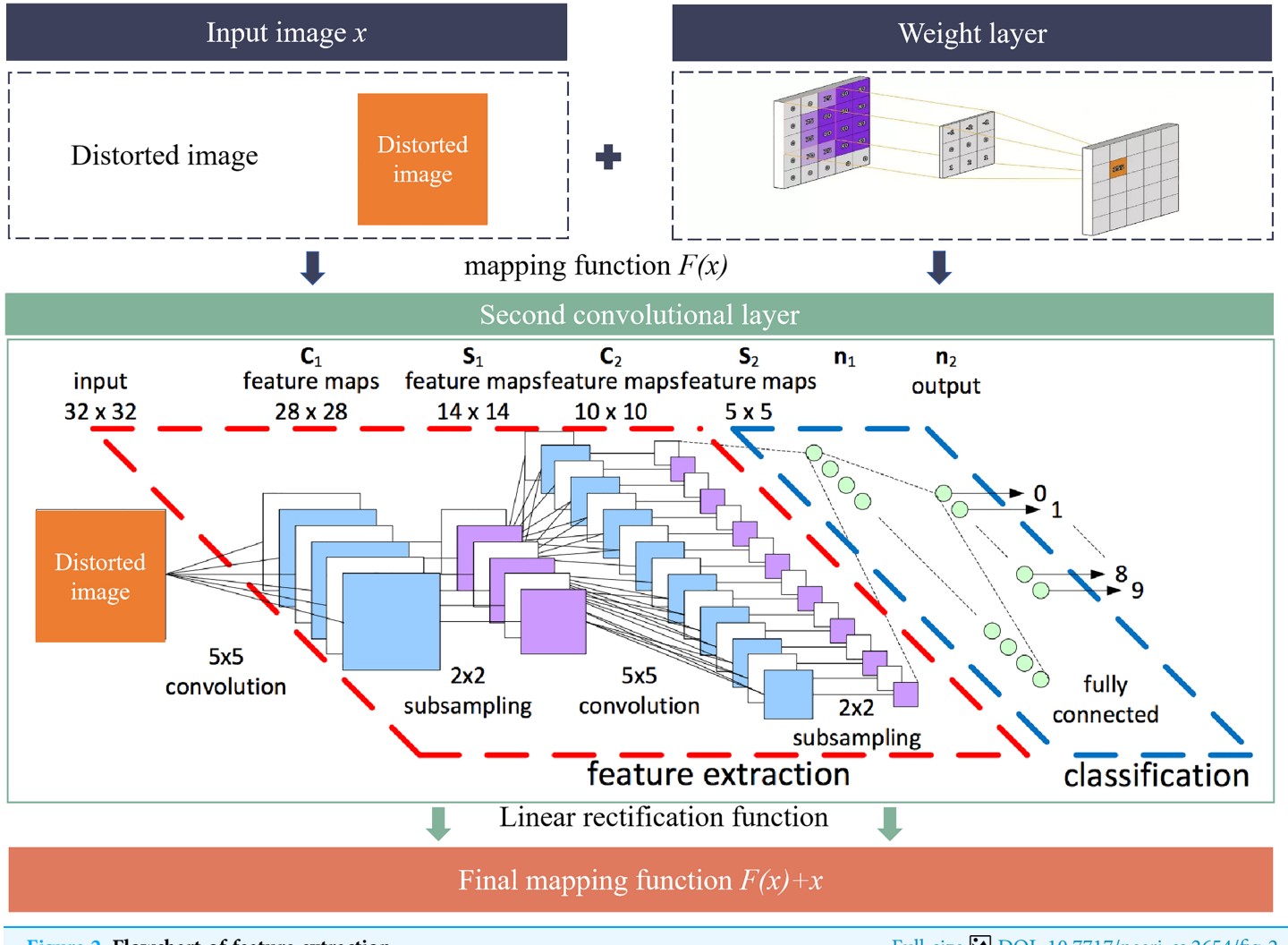

**Figure 2 Flowchart of feature extraction.**

function. The final output of the weight layer is added to the shallow information of the jump connection to obtain the final mapping function $F(x) + x$. This structure can directly map shallow information to deep positions, thus improving the accuracy of the model.

This article innovatively extracts the convolutional layer information $info_1$, $info_2$ and $info_3$ in the backbone network convolutional layer $c\_layer_{2\to10}$ (representing the tenth operation of the second convolutional layer), $c\_layer_{3\to12}$ and $c\_layer_{4\to18}$ as local semantic features of the image, and the convolutional layer information $info_4$ in $c\_layer_{5\to10}$ as the global semantic feature of the image. The method of obtaining multi-scale feature $info_i$ is shown in Eq. (1).

$$info_i = Res(x, \delta_1)(i = 1, \dots, 4).$$ (1)

Among them, $info_i$ represents the input image obtaining features of four different scales through the ResNet network, $Res()$ represents the feature extraction process of the

ResNet-50 network, $x$ represents the information of the input image, and $\delta_1$ represents the parameters of the ResNet-50 network. The fusion of multi-scale semantic features enables the complementary fusion of deep feature information and shallow feature information, which improves the accuracy of the model to a certain extent.

### Visual perception module

Because the traditional NR-IQA method based on deep learning network fails to fully incorporate the characteristics of the human visual perception system when designing the network structure, it leads to problems of insufficient sample number and limited sample diversity. This method often relies on limited data sets for training, which not only affects the generalization ability of the model, but also causes a large gap between the assessment results and human perception results. Therefore, to address these problems, this article innovatively introduces a reference-free image quality assessment method based on perceptual semantic features.

In this new approach, this article uses semantic features to describe the quality of images, and closely integrates the semantic content of images with quality assessment during the quality assessment process. In this way, not only can the human perception process of images be better simulated, but the automatic assessment process can also be made closer to human understanding and assessment of the semantic content of images. This improvement is expected to significantly improve the generalization ability of the model, making its performance on different samples more stable and reliable, and further narrowing the gap between automatic assessment results and human subjective perception. This article introduces a perception module to capture the information of distorted images. The calculation method of the perception module is shown in Eq. (2).

$$info'_i = per(info_i, \delta_2)(i = 1, \ldots, 4). \tag{2}$$

Among them, $info'_i$ represents the features of four different scales after being processed by the perception network model, $per()$ represents the perception network model, $info_i$ represents the features of four different scales obtained by the input image through the ResNet network, and $\delta_2$ represents the network parameters of the perception module. The network structure of this perception module is shown in Fig. 3.

Firstly, the local semantic features $info_i(i = 1, 2, 3)$ and the global semantic features $info_4$ output by the ResNet-50 network are divided into non-overlapping modules. Secondly, these non-overlapping modules are connected and stacked according to the number of dimensions of their respective channels. Finally, a 1 * 1 convolution operation and a global average pooling operation are performed to obtain the feature $info'_i(i = 1, \ldots, 4)$ processed by the perception module. Global average pooling calculates the mean of the two-dimensional feature map of each channel and takes the average of all pixel values of each channel as the output value of the channel. This operation greatly simplifies the representation of the data, thereby effectively reducing the number of parameters of the model. In this way, the complexity of the model is reduced and the risk of overfitting is effectively suppressed. In addition, this method can also preserve the

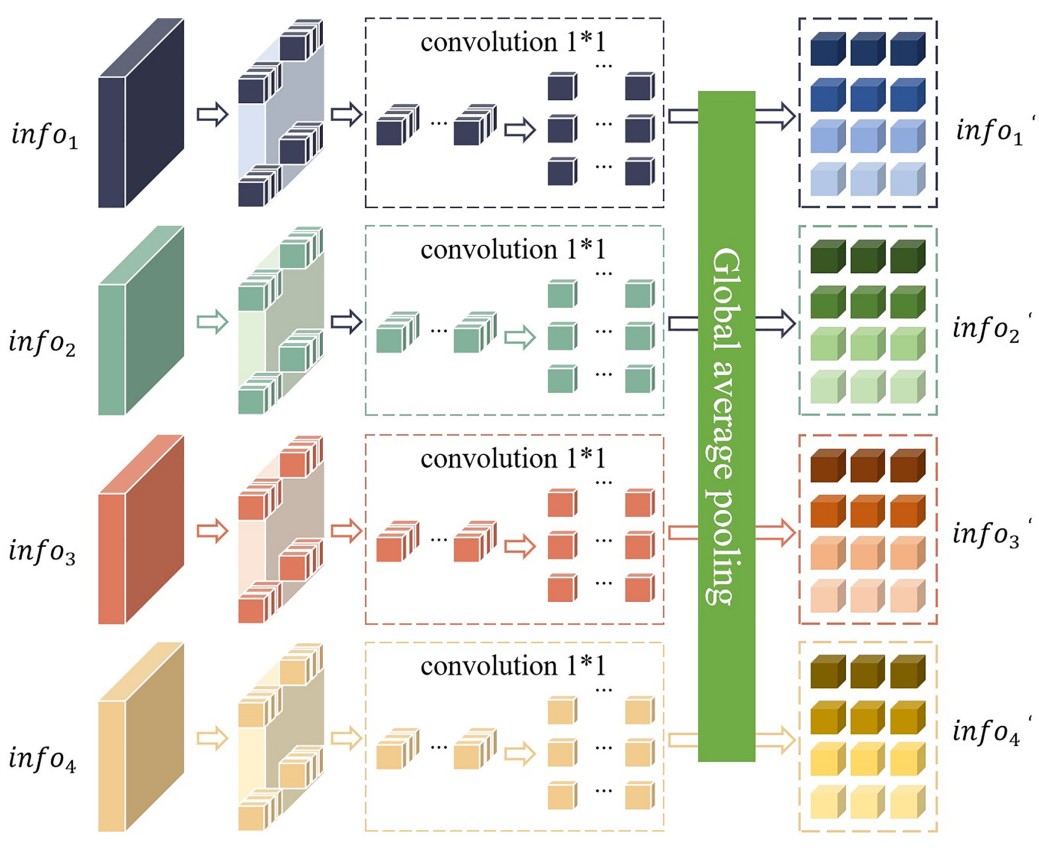

**Figure 3  Network structure of the perception module.**

spatial information of the input image, making the model more flexible and adaptable when processing inputs of different sizes.

### Adaptive fusion network module

In the feature fusion stage, a large amount of redundant information is usually inevitably introduced, which will interfere with the accurate assessment of image quality. In order to solve this problem and reduce the impact of noise superposition in different feature maps, this article applies an adaptive feature fusion module to the feature $info_i'(i = 1, 2, 3)$ processed by the perception module. The adaptive feature fusion module assigns higher weights to the key parts of each layer of features when performing feature fusion. Specifically, the adaptive fusion module can dynamically adjust the fusion ratio between different layers according to the importance of feature maps. This adaptive adjustment mechanism can not only highlight important feature information, but also effectively suppress unnecessary features, thereby optimizing the entire feature fusion process.

Through this approach, this article not only enhances the emphasis on important features, but also significantly reduces the amount of parameters in the network model. This reduction in the number of parameters further improves the efficiency of the model and makes the assessment of image quality more accurate and reliable. The network structure of this adaptive fusion network module is shown in Fig. 4.

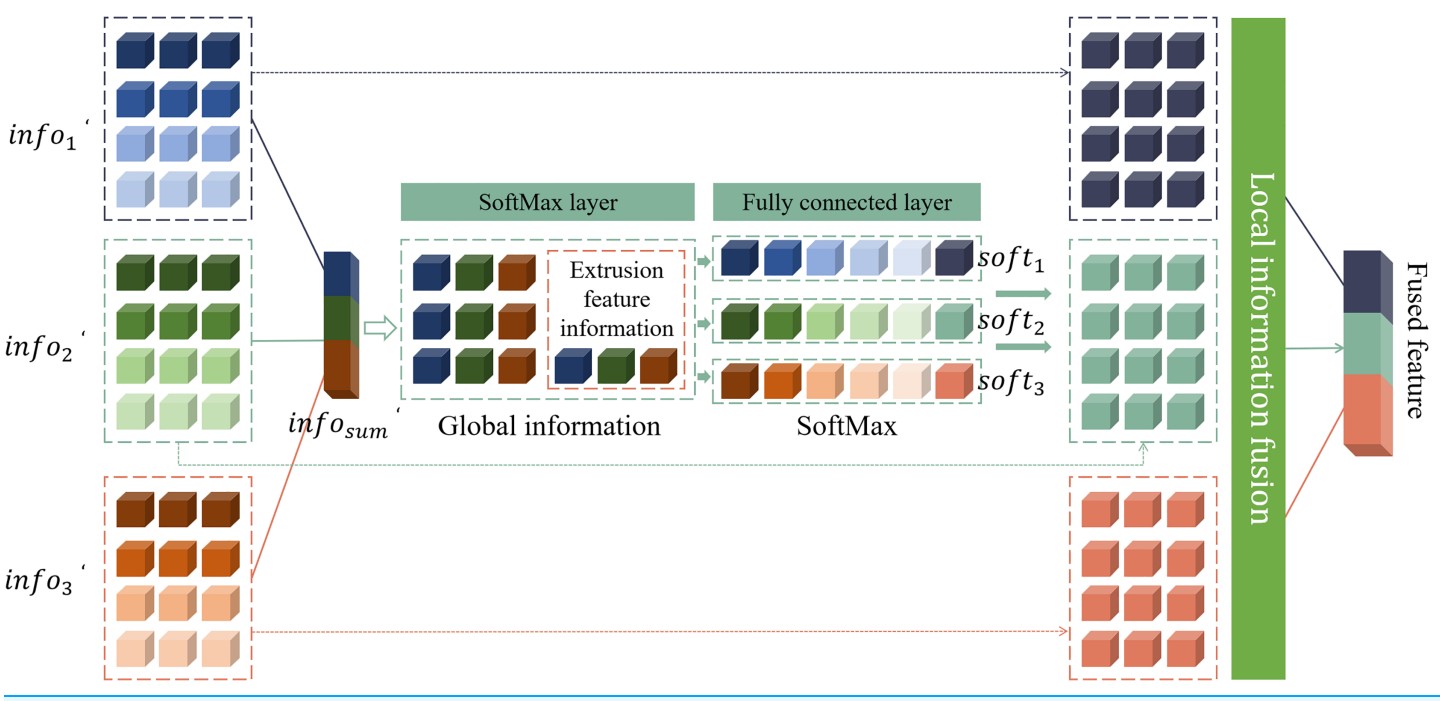

**Figure 4 Structure of the adaptive fusion network.**

The input of the network is the feature $info'_i (i = 1, 2, 3)$ processed by the perception module. These features have different dimensions, where $info'_1$ contains a 16-dimensional feature map, $info'_2$ contains a 32-dimensional feature map, and $info'_3$ contains a 64-dimensional feature map. Since the dimensions of these feature maps are inconsistent, in order to perform effective feature fusion in the model, this article converts them into feature maps of the same size. This model uniformly converts all feature maps into 64-dimensional feature maps. Specifically, this article uses 64 convolution kernels of size $1 \times 1$ to convolve these three features. The step size of each convolution kernel is different to ensure that different information in each feature map can be captured during the feature fusion process. In this way, this article can better fuse feature maps of different dimensions, so that the model can make full use of these features and improve the overall performance and accuracy. This feature fusion method not only ensures the consistency of feature map dimensions, but also can capture more local and global information through convolution operations of different step lengths, thereby enhancing the feature expression ability of the model. The features after dimension matching are summed up, as shown in Eq. (3).

$$info'_{sum} = M(info'_1) + M(info'_2) + M(info'_3). \tag{3}$$

Among them, $info'_{sum}$ represents the features obtained after fusion, $M()$ represents the dimension matching function, and $info'_i (i = 1, 2, 3)$ represents the features processed by the perception module. The size of the fused feature map is *height* ∗ *width* ∗ *channel* (where *height* represents the height of the feature map, *width* represents the width of the feature map, and *channel* represents the number of channels of the feature map). Then,

global average pooling is used to capture the global information of $info_i$. $info_i$ represents the features of four different scales obtained by the input image through the ResNet network, and the features are aggregated into a $channel * 1$-dimensional feature vector $info_{eig}$. Then, based on the fully connected layer, $info_{eig}$ is compressed into $(channel/coe) * 1$-dimensional squeezed feature information $info_{ext}$, where $coe$ is the compression coefficient, as shown in Eq. (4).

$$info_{ext} = A(matrix_{(channel/coe)*1} info_{eig}). \tag{4}$$

Among them, $info_{ext}$ represents the squeeze feature information, $A()$ represents the activation function, $matrix_{(channel/coe)*1}$ represents the $(channel/coe) * 1$ dimensional weighted matrix, and $info_{eig}$ represents the $channel * 1$ dimensional feature vector.

The model complexity can be reduced to a certain extent by the squeezing operation of the fully connected layer. Specifically, feature $info_{ext}$ is input to the $softmax$ layer and the fully connected layer, which can adaptively select important features from different scales. In this process, $soft_i(i = 1, 2, 3)$ represents the weight vector obtained through the adaptive network, which corresponds to feature selection at different scales. To be more specific, assume that $soft_i(i = 1, 2, 3)$ is the weight vector generated by the adaptive network at different scales, and $soft_i^j(i = 1, 2, 3)$ represents the $j - th$ element in $soft_i(i = 1, 2, 3)$, respectively. Due to the inherent properties of the $softmax$ layer, the elements of the weight vector $soft_i(i = 1, 2, 3)$ will satisfy the conditions of $\sum_{i=1}^{3} soft_i^j = 1$. This means that at each position $j$, the sum of the elements of the weight vector $soft_i(i = 1, 2, 3)$ is 1, ensuring the normalization of the weights and the reasonable distribution of relative importance. Finally, by operating the information with weights at different scales, the final feature map $f_{fused}$ can be obtained, as shown in Eq. (5).

$$f_{fused} = \sum_{i=1}^{3} soft_i info_i'. \tag{5}$$

Among them, $f_{fused}$ represents the final feature map, $soft_i(i = 1, 2, 3)$ represents the weight vector generated by the adaptive network at different scales, and $info_i'(i = 1, 2, 3)$ represents the features processed by the perception module. This method can not only effectively reduce the complexity of the model, but also improve the feature extraction ability and generalization performance of the model by adaptively selecting and combining important features at different scales.

### Quality assessment regression module

The target network of image quality assessment aims to accurately map the learned multi-scale image features to the corresponding quality scores. In this model structure, the target network used is composed of two fully connected layers. These two fully connected layers are responsible for fusing and processing information at different levels to ensure that the final output quality score can effectively reflect the quality of the input image. Specifically, the target network first receives the global semantic information $info_4$, which contains a high-level understanding of the overall characteristics of the input image. At the same

time, the target network also receives the feature $f_{fused}$ obtained through the adaptive fusion network. The role of the adaptive fusion network is to intelligently combine multi-scale image features to generate more representative feature representations. Then, through Eq. (6), the target network connects the global semantic information $info_4$ with the feature $f_{fused}$ to obtain the fused feature representation $f'_{fused}$. This process not only retains the global information of the image, but also integrates multi-scale features, so that the final $f'_{fused}$ can more comprehensively reflect the quality characteristics of the image. Finally, the target network maps $f'_{fused}$ to a specific quality score through further calculation, thus completing the image quality assessment task.

$$f'_{fused} = F(f_{fused}, info'_4). \tag{6}$$

Among them, $f'_{fused}$ represents the fused features, $F()$ represents the feature fusion function, $f_{fused}$ represents the features obtained by the adaptive fusion network, and $info'_4$ represents the features processed by the perception module.

After the obtained feature $f'_{fused}$ is used as the input of the target network, it is further processed. This article selects the *LeakyReLU* function as the activation function of the target network. The *LeakyReLU* function is a commonly used activation function that allows a certain negative slope when the input value is negative, thereby alleviating the "neuron death" problem that may occur in the *ReLU* function to a certain extent. This target network inputs the feature $f'_{fused}$ into fully connected layers activated by the *LeakyReLU* activation function, which perform nonlinear transformations on the input features to capture important patterns and features required in image quality assessment. Finally, after multiple layers of processing, the target network outputs an image quality score, which represents a comprehensive assessment result of the input image quality. This quality score is shown in Eq. (7), which specifically describes the calculation process from feature $f'_{fused}$ to the final quality score. Through this process, the network can effectively use the learned features to evaluate the image quality and return a quantitative quality score S.

$$S = Eva(f'_{fused}, weight_{full}, bias). \tag{7}$$

Among them, S represents the final predicted image quality score, $Eva()$ represents the quality assessment network model, $f'_{fused}$ represents the fused features, $weight_{full}$ represents the fully connected layer weight of the target network, and CC represents the fully connected layer *bias* of the target network.

Based on the above content, the overall network model proposed in this article is shown as Eq. (8).

$$S_{final} = T\_Eva(x, info_i, info'_i, S)(i = 1, 2, 3) \tag{8}$$

Among them, $S_{final}$ represents the final quality score of the image, $T\_Eva()$ represents the entire network model, $x$ represents the input distorted image, $info_i$ represents the multi-scale feature extraction network model from ResNet-50, $info'_i$ represents the perception network model, S represents the quality assessment network model, and $i$

represents the features of $i$ different scales after processing by the perception network model.

# EXPERIMENTS AND ANALYSES

## Datasets

To ensure the objectivity and effectiveness of the experiments, this article utilizes a combination of real-world and artificially synthesized distortion datasets, including KonIQ-10K (*Hosu et al., 2020*), LIVE (*Edlund et al., 2021*), BIQ2021 (*Ahmed & Asif, 2022*), and TID2013 (*Ponomarenko et al., 2015*).

Among these, KonIQ-10K is a widely adopted dataset for image quality assessment. It comprises 10,073 real-world images collected from a variety of online platforms, representing diverse shooting conditions and devices. Each image in the dataset is associated with multiple subjective quality ratings provided by independent human reviewers, making it a valuable resource for understanding perceptual image quality in natural scenes.

The LIVE dataset, created by the University of Texas, is a well-established benchmark in the field. It contains a broad range of images and video clips subjected to common distortion types, such as compression artifacts, Gaussian noise, blur, and transmission errors. The subjective scores for each image and video are derived from evaluations by human observers, providing valuable insights into how these distortions affect perceived quality.

In addition to these, the BIQ2021 dataset plays a critical role in NR-IQA tasks. Designed to handle real-world distortions, BIQ2021 includes a diverse collection of both natural and distorted images. The dataset provides subjective scores gathered from human observers, facilitating research focused on blind or no-reference IQA. Its diverse range of real-world distortions makes it an essential dataset for evaluating modern NR-IQA methods.

Lastly, the TID2013 dataset is one of the most comprehensive resources available for image quality assessment, encompassing 25 reference images with 24 types of distortions at five levels of intensity. These distortions cover a wide range of visual anomalies, including Gaussian noise, image compression, chromatic aberrations, and more. Each image is accompanied by subjective quality ratings collected from human participants, allowing for detailed analysis of how different distortions impact perceptual quality. The richness of the TID2013 dataset makes it an indispensable resource for both full-reference and no-reference IQA model evaluation.

The number of datasets in the field of image quality assessment is relatively small, and it is difficult to find enough data to complete experiments. This makes data preprocessing operations particularly important in quality assessment. In order to increase the training data of the model, this experiment uses data augmentation. First, for the training data in the *LIVE* dataset, this article performs random sampling operations to obtain different image samples. Then, these samples are horizontally inverted and the image size is adjusted to 224 ∗ 224 pixels. Next, data format conversion and data standardization are performed to ensure that the data has a consistent format and distribution before entering

the model. For the $KonIQ - 10K$ dataset, this article first resizes the training data to $512 * 384$ pixels. Then, it is processed according to the same steps as the $LIVE$ dataset, including random sampling, horizontal reversal, data format conversion, and data normalization. These preprocessing operations not only increase the amount of data, but also enrich the diversity of the data.

### Dataset introduction and data augmentation

Due to the limited availability of datasets in the field of image quality assessment, acquiring sufficient experimental data can be challenging.

1. LIVE

This dataset contains numerous images and videos with a variety of distortions, including compression artifacts, noise, blur, and transmission errors. Like the KonIQ-10K dataset, each image and video in the LIVE dataset is rated subjectively by human observers. For the LIVE dataset, the process includes:

- Random Sampling: Selecting various image samples to increase data variability.
- Horizontal Flipping: Enhancing data diversity through mirroring.
- Resizing: Adjusting images to a uniform size of $224 \times 224$ pixels.
- Format Conversion and Normalization: Ensuring consistency in data format and distribution.

2. TID2013

The TID2013 dataset is one of the most comprehensive datasets used for image quality assessment, containing images with 24 types of distortions at five different levels of severity. For the TID2013 dataset, the process includes:

- Resizing: All images are resized to $384 \times 384$ pixels to ensure uniformity across the dataset.
- Data Augmentation: Techniques such as random cropping and flipping are applied to increase data variability and model performance.
- Format Conversion and Normalization: Consistent with other datasets, this step ensures that all images are in the same format and normalized to a standard distribution.

3. KonIQ-10K

This dataset is widely utilized for image quality evaluation. It includes over 10,000 images gathered from various online platforms, covering a range of shooting conditions and devices. For the KonIQ-10K dataset, the process includes:

- Resizing: Adjusting images to $512 \times 384$ pixels.
- Random Sampling and Horizontal Flipping: Similar to the LIVE dataset, to enrich the dataset.
- Data Format Conversion and Normalization: Consistent with the preprocessing steps used for the LIVE dataset.

4. BIQ2021

The BIQ2021 dataset is a more recent addition to the field of image quality assessment, specifically focusing on blind or no-reference image quality tasks. It includes a diverse range of images, featuring both natural and distorted content. For the BIQ2021 dataset, the process includes:

- Resizing: Images are resized to $256 \times 256$ pixels to maintain consistency.
- Random cropping and horizontal flipping: These techniques are used to augment the dataset and improve model robustness.
- Format conversion and normalization: Ensures consistency in data format and pixel value distribution across all samples.

The code used in this experiment has been made publicly available on GitHub. You can access it *via* the following link: https://github.com/deyi2024/A-Quality-Assessment-Algorithm-for-No-Reference-Images-Based-on-Transfer-Learning.git.

### *Dataset with manual scoring*

Unlike NSIs, which are captured from real scenes, AGIs are produced by AI models and exhibit unique quality characteristics. Consequently, viewers assess AGIs differently from NSIs. To address this, *Zhang et al. (2023c)* introduced the AGIQA-1K database, which features 1,080 AGIs with quality labels covering technical issues, AI artifacts, unnaturalness, discrepancy, and aesthetics.

### Algorithm effect assessment indicators

This algorithm uses the two most commonly used model performance assessment indicators in the field of image quality assessment to comprehensively measure the effect of a no-reference image quality assessment algorithm based on transfer learning. The two indicators are Pearson linear correlation coefficient (PLCC) and Spearman rank correlation coefficient (SROCC). PLCC is a statistic used to measure the strength of the linear relationship between two variables. In image quality assessment, PLCC is used to evaluate the linear correlation between the quality score predicted by the model and the subjective score of humans. A higher PLCC value indicates that there is a strong linear relationship between the quality score predicted by the model and the subjective score, which means that the model can accurately reflect human perception of image quality. The algorithm proposed in this article calculates PLCC as shown in Eq. (9).

$$S_{PLCC} = \frac{\sum_{i=1}^{n}(real_i - \overline{real})(pred_i - \overline{pred})}{\sqrt{\sum_{i=1}^{n}(real_i - \overline{real})^2}\sqrt{\sum_{i=1}^{n}(pred_i - \overline{pred})^2}}. \tag{9}$$

Among them, $S_{PLCC}$ represents the Pearson linear correlation coefficient value, $i$ represents the $i - th$ distorted image, $n$ represents the total number of distorted images, $real_i$ represents the subjective assessment score of the distorted image, $\overline{real}$ represents the mean of the subjective score, $pred_i$ represents the model prediction score of the distorted image, and $\overline{pred}$ represents the mean of the prediction score.

*PLCC* is a commonly used indicator to describe the linear correlation between two sets of data, and its value range is between [−1, 1]. Specifically, when the value of *PLCC* is 0, it means that the two sets of data are completely unrelated, that is, there is no linear relationship between the quality score predicted by the model and the subjective assessment of humans. The closer the *PLCC* value is to 1, the stronger the positive linear relationship between the two sets of data. In other words, the closer the model-predicted score is to the standard of human subjective assessment, the better the model's performance. On the contrary, if the *PLCC* value is close to −1, it means that there is a strong negative linear relationship between the two sets of data, which means that the model's prediction results are exactly the opposite of the trend of human subjective assessment. In this case, the model's prediction performance is poor and cannot effectively reflect humans' actual perception of image quality.

*SROCC* is a nonparametric statistic that measures the order relationship between two variables. *SROCC* is used in image quality assessment to evaluate the monotonic relationship between model prediction results and subjective scores, that is, whether the two variables have a consistent ranking trend. A higher *SROCC* value indicates that the ranking of the model prediction results is more consistent with the ranking of the subjective scores, which means that the model can maintain a good relative order when predicting image quality. These two performance assessment indicators can comprehensively measure the prediction accuracy and monotonicity of the prediction results of the algorithm proposed in this article. The *SROCC* calculated by the algorithm proposed in this article is shown in Eq. (10).

$$S_{SROCC} = 1 - \frac{6 \sum_{i=1}^{n} grade_i^2}{n(n^2 - 1)}. \tag{10}$$

Among them, $S_{SROCC}$ represents the Spearman rank correlation coefficient value, $i$ represents the $i - th$ distorted image, $n$ represents the total number of distorted images, and $grade_i$ represents the grade difference between the predicted score and the subjective score of the distorted image.

*SROCC* is often used to represent the rank correlation between two sets of data, and its value range is [0, 1]. When the *SROCC* value is closer to 1, it means that the consistency between the two sets of data is higher. In image quality assessment, a higher *SROCC* value indicates that there is a high degree of consistency between the quality score predicted by the model and the ranking of human subjective assessment, indicating that the model can well imitate human quality perception ability when predicting image quality, indicating that the model's predicted score is very close to the subjective quality assessment.

## Comparison of experimental results
### Comparison based on the real distortion dataset *KonIQ − 10K* **and** *BIQ*2021
The research method in this article was compared with the assessment methods proposed by other scholars on the same datasets (*KonIQ − 10K* and *BIQ*2021). First, 90% of all target images in the dataset are randomly selected as training data, and the remaining 10%

are used as test data. In order to ensure the robustness of the experimental results and reduce the accidental effects caused by data division, the division of the training set and the test set is repeated 50 times, and the median of all results is taken as the final experimental result. In order to reduce the impact of outliers on model training, the mean absolute error (MAE) was used as the loss function during the training process. MAE can minimize the gap between the model prediction results and the actual subjective quality score, making the model's prediction results as close to the subjective quality score as possible. The calculation method of the loss function is shown in Eq. (11). By minimizing this loss function, the model can more accurately reflect the true quality of the image.

$$Loss = \frac{\sum_{i=1}^{Sum} ||T\_Eva(piece_i, info_i, info'_i, S) - Q\_real_i||}{Sum}. \tag{11}$$

Among them, *Loss* represents the loss function value, *Sum* represents the total number of image blocks, $T\_Eva()$ represents the entire network model, $piece_i$ represents the $i - th$ image block of the input distorted image, $info_i$ represents the multi-scale feature extraction network model from $ResNet - 50$, $info'_i$ represents the perception network model, $S$ represents the quality assessment network model, and $Q\_real_i$ represents the true quality score of the $i - th$ image block.

This experiment compares the image quality assessment methods based on manually extracted image features, such as the *IQA—NRTL* method proposed in this article, the *ECGQA* algorithm proposed by *Liu et al. (2021)*, the *DLBF* algorithm proposed by *Mahum & Aladhadh (2022)*, the *FDD* algorithm proposed by *Chen, Rottensteiner & Heipke (2021)*, and the image quality assessment methods based on deep learning, such as the *CELL* algorithm proposed by *Rasheed, Shi & Khan (2023)*, the *PDAA* algorithm proposed by *Valicharla et al. (2023)*, and the *CFFA* algorithm proposed by *König et al. (2024)*. The experiments are carried out on the authentically distorted images from $KonIQ - 10K$ and $BIQ2021$ dataset, and the algorithm effects are compared with the *PLCC* and *SROCC* values, as shown in Table 1.

As can be seen from Table 1, the assessment methods based on manual feature extraction, such as *ECGQA*, *DLBF*, and *FDD*, perform poorly on real distortion datasets, which is specifically reflected in the low *PLCC* and *SROCC* values and the inability to accurately predict the image quality score. This shows that these methods have obvious limitations when dealing with complex image quality assessment tasks and cannot effectively capture the true quality characteristics of images. In contrast, deep learning-based assessment methods such as *CELL*, *PDAA*, and *CFFA* perform much better on real distortion datasets. These methods learn image features through deep neural networks and improve the accuracy of image quality prediction to a certain extent. However, these methods still have difficulty in achieving ideal prediction accuracy for various reasons. The *CELL* method lacks a module that captures local semantic information, which makes it unable to fully understand the local quality changes of the image; although the *PDAA* method considers the importance of local information for global quality prediction, it ignores the correlation between the extracted hierarchical features, which limits its

**Table 1 Comparative experimental results based on the *KonIQ − 10K* and *BIQ*2021 dataset.**

| Comparison algorithms | | KonIQ-10K results | | BIQ2021 results | |
|---|---|---|---|---|---|
| | | PLCC | SROCC | PLCC | SROCC |
| Based on hand-crafted statistics | ECGQA | 0.650 | 0.549 | 0.703 | 0.612 |
| | DLBF | 0.694 | 0.614 | 0.743 | 0.679 |
| | FDD | 0.755 | 0.498 | 0.802 | 0.564 |
| Based on deep learning | CELL | 0.713 | 0.710 | 0.768 | 0.766 |
| | PDAA | 0.782 | 0.638 | 0.831 | 0.693 |
| | CFFA | 0.669 | 0.802 | 0.727 | 0.865 |
| IQA-NRTL | | 0.895 | 0.850 | 0.938 | 0.902 |

performance improvement; the adaptive layer of the *CFFA* method ignores important underlying distortion information when selecting features, making its prediction results imperfect.

The *IQA—NRTL* method proposed in this article improves the above shortcomings. *IQA—NRTL* is designed to pay more attention to the comprehensiveness and accuracy of feature extraction, especially in capturing local semantic information and processing multi-layer feature correlation. Experimental results show that the *PLCC* and *SROCC* values of the *IQA—NRTL* method on the real distortion data set have reached the optimal level, significantly better than traditional manual feature extraction methods and existing deep learning methods. Among them, in terms of *PLCC*, the *IQA—NRTL* method improves by 15.64% and 12.68% respectively compared with the optimal solutions of the first two strategies. In terms of *SROCC*, the *IQA—NRTL* method improves by 27.76% and 5.65% respectively compared with the optimal solutions of the first two strategies. This shows that the *IQA—NRTL* method has high accuracy and reliability in evaluating data sets of real natural scenes.

### Comparison based on artificial distortion dataset LIVE and *TID*2013

In order to verify the performance of this method on a synthetic dataset, this article selected the *LIVE* and *TID*2013 dataset, which is widely used in the field of image quality assessment. Through this dataset, this article systematically compared and analyzed the proposed assessment method with other commonly used quality assessment methods. Specifically, the *PLCC* and *SROCC* values of each method on the *LIVE* dataset were calculated to measure the accuracy and consistency of different methods in image quality prediction. Table 2 shows the *PLCC* and *SROCC* values of each method on the *LIVE* data set. It can be clearly seen from the results that the *IQA—NRTL* method proposed in this article performs well on both key performance indicators, significantly better than traditional manual feature extraction methods and existing deep learning methods.

From the data in Table 2, it can be concluded that the assessment method *IQA—NRTL* proposed in this article has achieved relatively ideal test results on the *LIVE* dataset, and its *PLCC* and *SROCC* values are better than most methods. Although the assessment score of

**Table 2 Comparative experimental results based on the *LIVE* and *TID*2013 dataset.**

| Comparison algorithms | | LIVE results | | TID2013 results | |
|---|---|---|---|---|---|
| | | PLCC | SROCC | PLCC | SROCC |
| Based on hand-crafted statistics | ECGQA | 0.908 | 0.859 | 0.853 | 0.802 |
| | DLBF | 0.915 | 0.848 | 0.862 | 0.789 |
| | FDD | 0.906 | 0.901 | 0.851 | 0.860 |
| Based on deep learning | CELL | 0.912 | 0.910 | 0.860 | 0.857 |
| | PDAA | 0.920 | 0.930 | 0.871 | 0.883 |
| | CFFA | 0.877 | 0.817 | 0.821 | 0.764 |
| IQA-NRTL | | 0.914 | 0.929 | 0.870 | 0.874 |

the *IQA—NRTL* method did not reach the best among all mainstream image quality assessment methods, its *PLCC* index reached 0.914, which was only 0.006 lower than the best algorithm, and its *SROCC* index reached 0.929, which was only 0.009 lower than the best method. This shows that the *IQA—NRTL* method can provide reliable quality prediction results in practical applications while maintaining high accuracy and consistency. Compared with the mainstream *FDD* and *PDAA* methods, the performance of the *IQA—NRTL* method in practical applications is not substantially different, and it can meet the needs of actual image quality assessment and achieve ideal quality prediction results. In summary, by comparing the experimental results, it can be seen that the *IQA—NRTL* method proposed in this article performs quite well on the *LIVE* dataset. Although it is slightly inferior to the best method in some indicators, its overall performance is comparable to the current mainstream image quality assessment methods.

### Fusion experiment

To further verify the effectiveness of the *IQA—NRTL* method proposed in this article, this article conducted multiple comparative experiments on the *KonIQ* − 10*K*, *BIQ*2021, *LIVE* and *TID*2013 dataset. First, this article uses a model containing only the *ResNet* − 50 network as the baseline method for experiments, named *Exp_ResNet* − 50; then, the multi-scale semantic information feature extraction module, visual perception module and adaptive fusion network module are added in turn for experiments, named *Exp_Semantic*, *Exp_Vision* and *Exp_Adapt* respectively; finally, the *IQA—NRTL* model containing all modules is used for experiments. Through these experiments, the experimental results of the five models were obtained and their *PLCC* and *SROCC* values were compared. The experimental results are shown in Table 3.

Through the above experimental steps, we can compare in detail the effect of each module on improving model performance, and finally verify the effectiveness and superiority of the *IQA—NRTL* method proposed in this article in image quality assessment. Experimental results show that the addition of each module significantly improves the performance of the model, and the final *IQA—NRTL* model performs best in both *PLCC* and *SROCC* indicators.

**Table 3 Comparative experimental results.**

| Comparison algorithms | KonIQ-10K | | BIQ2021 | | LIVE | | TID2013 | |
|---|---|---|---|---|---|---|---|---|
| | PLCC | SROCC | PLCC | SROCC | PLCC | SROCC | PLCC | SROCC |
| Exp_ResNet-50 | 0.815 | 0.721 | 0.857 | 0.762 | 0.912 | 0.918 | 0.883 | 0.881 |
| Exp_Semantic | 0.826 | 0.810 | 0.868 | 0.852 | 0.927 | 0.937 | 0.892 | 0.902 |
| Exp_Vision | 0.819 | 0.799 | 0.852 | 0.849 | 0.924 | 0.928 | 0.894 | 0.891 |
| Exp_Adapt | 0.834 | 0.749 | 0.871 | 0.784 | 0.928 | 0.936 | 0.891 | 0.906 |
| IQA-NRTL | 0.887 | 0.859 | 0.928 | 0.891 | 0.949 | 0.955 | 0.912 | 0.928 |

**Table 4 Comparative experimental results based on the AGIQA-1K dataset.**

| Comparison algorithms | | Algorithm results | | AGIQA-1K results | |
|---|---|---|---|---|---|
| | | PLCC | SROCC | PLCC | SROCC |
| Based on hand-crafted statistics | ECGQA | 0.894 | 0.827 | 0.846 | 0.783 |
| | DLBF | 0.870 | 0.786 | 0.836 | 0.793 |
| | FDD | 0.875 | 0.828 | 0.827 | 0.826 |
| Based on deep learning | CELL | 0.901 | 0.899 | 0.825 | 0.893 |
| | PDAA | 0.912 | 0.927 | 0.804 | 0.810 |
| | CFFA | 0.863 | 0.801 | 0.736 | 0.762 |
| IQA-NRTL | | 0.909 | 0.921 | 0.903 | 0.910 |

***Subjective experiment based on** AGIQA − 1K*

Both the algorithms based on "Hand-Crafted Statistics" and "Deep Learning," as well as the IQA-NRTL algorithm proposed in this article, achieved satisfactory results in image quality assessment experiments, as shown in Table 4. However, in subjective evaluation experiments, the results of the IQA-NRTL algorithm were closer to human visual scores compared to other algorithms. This indicates that the proposed algorithm demonstrates a clear advantage in subjective evaluations, further underscoring its ability to better simulate the human visual system's perception of image quality. This advantage highlights the superior practical reliability of the IQA-NRTL algorithm in image quality assessment tasks.

## CONCLUSION

This article presents an advanced no-reference image quality assessment method, IQA-NRTL, based on deep learning that leverages transfer learning combined with a convolutional neural network to address common challenges in image quality assessment. By integrating transfer learning, IQA-NRTL increases the available training data, mitigating overfitting and enhancing generalization. The design incorporates a multi-scale information extraction approach, enabling the model to capture distortion characteristics at various levels and ensuring effective identification of quality issues. Additionally, a perception module refines semantic information extraction by reducing the impact of

irrelevant details, while an adaptive fusion module integrates global and local features, improving the accuracy and comprehensiveness of quality assessments. A cognitive memory library through a fully connected layer regression network provides robust predictions for distorted images. Experimental results validate the effectiveness of IQA-NRTL, demonstrating a 15.64% improvement in PLCC and a 27.76% improvement in SROCC compared to the best traditional methods, confirming its superiority in image quality assessment.

### Funding
This article is supported by the National Social Science Fund of China (Grant no. 22BSH025), the National Natural Science Foundation of China (Grant no. 62206241) and the Key Research and Development Program of Zhejiang Province, China (grant no. 2021C03138), and the Medium and Long-Term Science and Technology Plan for radio, television, and online audiovisuals (Grant no. 2022AD0400). The funders had no role in study design, data collection and analysis, decision to publish, or preparation of the manuscript.

### Grant Disclosures
The following grant information was disclosed by the authors:
National Social Science Fund of China: 22BSH025.
National Natural Science Foundation of China: 62206241.
Key Research and Development Program of Zhejiang Province, China: 2021C03138.
Medium and Long-Term Science and Technology Plan for radio, television, and online audiovisuals: 2022AD0400.

### Competing Interests
The authors declare that they have no competing interests.

### Author Contributions
- Yang Yang conceived and designed the experiments, analyzed the data, performed the computation work, prepared figures and/or tables, authored or reviewed drafts of the article, and approved the final draft.
- Chang Liu conceived and designed the experiments, performed the experiments, performed the computation work, prepared figures and/or tables, authored or reviewed drafts of the article, and approved the final draft.
- Hui Wu conceived and designed the experiments, performed the experiments, analyzed the data, prepared figures and/or tables, authored or reviewed drafts of the article, and approved the final draft.
- Dingguo Yu performed the experiments, analyzed the data, performed the computation work, authored or reviewed drafts of the article, and approved the final draft.

## Data Availability

The third party datasets are available at:

- KonIQ-10K:

https://database.mmsp-kn.de/koniq-10k-database.html

- LIVE:

https://utexas.box.com/v/ChallengeDB-release

- BIQ2021: Nisar Ahmed. (2023). BIQ2021: A Dataset for Image Quality Assessment [Data set]. GitHub. https://github.com/nisarahmedrana/BIQ2021

- TID2013: https://www.ponomarenko.info/tid2013.htm

The code is available at Zenodo: John. (2024). A Quality Assessment Algorithm for No-Reference Images Based on Transfer Learning. Zenodo. https://doi.org/10.5281/zenodo.13423938.

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
