# Peer review of "A quality assessment algorithm for no-reference images based on transfer learning"

_PeerJ Computer Science, doi:10.7717/peerj-cs.2654_

## Round 0.1 · original submission · Major Revisions

Dear Author,

Your manuscript needs revisions before it can be accepted.

Check carefully the comment of the reviewers and address ALL of them.

Best,

M.P.

·

Basic reporting

Thank you for submitting your original paper “A Quality Assessment Algorithm for No-Reference Images based on Transfer Learning”. In this time, there are some questions and comments.

(a). In Abstract, you should explain why you used the deep convolutional neural network in the case of considering the image quality assessment.
(b). In Abstract, “The experimental results of the paper show that compared with the mainstream image quality assessment algorithms, the IQA-NRTL algorithm proposed in the paper improves the effect of no-reference image quality assessment by 5.65%-27.76%.” 5.65%-27.76% is too large range. You should add explanation for this result.
(c). In Introduction, you should clarify purpose of research in this study. You should explain why carrying out the image quality assessment.
(d). You should explain what meaning and goal is algorithm which you showed.
(e). You should clarify contents in datasets. Contents are importance to assess image quality. By changing contents characteristics, assessment value also is changed.
In this paper, you consider the only objective quality assessment. You should also consider subjective quality assessment of your proposed method.

Experimental design

You should carry out not only the objective quality assessment but also the subjective quality assessment.

Validity of the findings

You should add contents of discussion for experimental results.

Additional comments

no comment.

·

Basic reporting

The introduction section of the paper requires significant revision. The background information provided is not relevant to the study's focus, making it unclear why it was included. Additionally, the sections under “Image Quality Assessment” and “No-reference Image Quality Assessment” also need to be rewritten to better align with the standard structure of a research article introduction.

Typically, the introduction should include a clear background and context, followed by a well-defined problem statement. It is also common to include research questions or objectives, the scope of the study, and the significance of the results or contributions. However, these elements are largely missing in the current version of the paper.

Moreover, the paper would benefit from a separate section dedicated to literature review or related work. This section should discuss existing studies that utilize transfer learning or deep feature extraction for image quality assessment and clearly highlight the research gap addressed by this study.

Experimental design

Although the study falls within the scope of the journal, it lacks a comprehensive review of related work and a clear articulation of the research gap. Additionally, the approach presented is similar to those used in existing studies and, as a result, does not offer a significant contribution to the field.

Regarding the experiments, the paper focuses on the KonIQ-10k and LIVE datasets, but it overlooks other relevant and recent datasets like BIQ2021 and TID2013. It would strengthen the study to include experiments using these datasets, particularly by testing authentically distorted images with KonIQ-10k and BIQ2021, and synthetically distorted images with TID2013 and LIVE.

The comparative analysis with existing approaches is insufficient, as it omits many recent methods that have performed well on the mentioned datasets. The authors should refer to the recent paper by Aslam et al. (IEEE Access, 2024) to identify related studies and perform a more comprehensive comparison.

The section titled "ALGORITHM FLOW AND IMPLEMENTATION STEPS" could be better organized and presented as a methods section, consistent with other papers in the domain. Furthermore, an ablation study should be included to demonstrate the contribution of various model components to the overall performance.

Reference:
M. A. Aslam, X. Wei, N. Ahmed, G. Saleem, T. Amin and H. Caixue, "VRL-IQA: Visual Representation Learning for Image Quality Assessment," in IEEE Access, vol. 12, pp. 2458-2473, 2024, doi: 10.1109/ACCESS.2023.3340266.

Validity of the findings

The results reported in the paper are not sufficient. The authors need to conduct an ablation study and provide a more thorough comparison with existing studies, as mentioned in the “Experimental Design” section of this review.

Additional comments

The citations in the paper are not appropriately formatted. Proper citation formatting is essential for clarity and credibility, so these should be corrected to align with the journal's guidelines.

---

## Round 0.2 · Minor Revisions

Dear Authors,

Please take into account the final comments related to rewriting the introduction and literature review sections.

·

Basic reporting

The first paragraph of Introduction section of the manuscript directly starts with AI generated content (AIGC) which doesn't align well with the manuscript. The title of the manuscript highlight that the paper propose an image quality assessment algorithm for no-reference images and therefore the introduction should start with background of this fundamental problem.

The authors are advised to update the introduction section through rewriting it and presenting the information in an appropriate manner. Moreover, the abstract of the manuscript can mention the datasets which are used to conduct quality evaluation experiments and the last sentence can be written as "Experimental results on authentically distorted datasets (KonIQ-10k & BIQ2021), synthetically distorted datasets (LIVE & TID2013) and AI generated content dataset (AGIQA-1K) show that the proposed IQA-NRTL algorithm significantly improves performance, ... etc"
This change will highlight the contribution of the proposed approach and guide the reader that the proposed approach is extensively evaluated on different types of datasets.

The subsection "Purpose of the Research" in Literature Review section is not appropriately placed and can be removed and the information in this section can be accommodated in introduction section while rewriting it.

Experimental design

The experimental section is improved as per the suggestions provided in earlier review and is in satisfactory condition.

Validity of the findings

To enhance the reproducibility and credibility of the results, it is recommended that the authors share the code or provide access to a GitHub repository containing their implementation. This will facilitate validation and help others replicate the findings more easily.

Additional comments

The authors are requested to get the final manuscript reviewed to improve the readability of the content as some of the sentences are written in a complex structure which impairs the readability.

---

## Round 0.3 · Minor Revisions

Dear Authors,

you addressed all the reviewer's comments.

I STRONGLY suggest to upload your code in a public repository (e.g. GitHub) and share with the community for reproducibility and transparency. Please, put the link to the repository in the paper.

M.P.

·

Basic reporting

The manuscript has addressed the highlighted concerns and is therefore recommended for acceptance.

Experimental design

The concern are already addressed.

Validity of the findings

The authors has yet to upload the code to GitHub or some other repository for transparency.

Additional comments

No comments.

---

## Round 0.4 · accepted · Accept

Dear Authors,

You have addressed all of the reviewers' comments and the manuscript is ready for publication.

M.P.